# Microbial Diversity Impacts Non-Protein Amino Acid Production in Cyanobacterial Bloom Cultures Collected from Lake Winnipeg

**DOI:** 10.3390/toxins16040169

**Published:** 2024-03-26

**Authors:** Stephanie L. Bishop, Julia T. Solonenka, Ryland T. Giebelhaus, David T. R. Bakker, Isaac T. S. Li, Susan J. Murch

**Affiliations:** 1Department of Chemistry, University of British Columbia, Syilx Okanagan Nation Territory, Kelowna, BC V1V 1V7, Canada; jtsolone@student.ubc.ca (J.T.S.); rgiebelh@ualberta.ca (R.T.G.); davidtrb@live.ca (D.T.R.B.); isaac.li@ubc.ca (I.T.S.L.); susan.murch@ubc.ca (S.J.M.); 2Department of Biological Sciences, University of Calgary, Calgary, AB T2N 1N4, Canada; 3Department of Chemistry, University of Alberta, Edmonton, AB T6G 2N4, Canada; 4The Metabolomics Innovation Centre, Edmonton, AB T6G 2N4, Canada

**Keywords:** β-*N*-methylamino-L-alanine (BMAA), *N*-(2-aminoethyl)glycine (AEG), 2,4-diaminobutyric acid (DAB), non-protein amino acids (NPAAs), harmful algal blooms, Lake Winnipeg, Canada, cyanotoxin production

## Abstract

Lake Winnipeg in Manitoba, Canada is heavily impacted by harmful algal blooms that contain non-protein amino acids (NPAAs) produced by cyanobacteria: *N*-(2-aminoethyl)glycine (AEG), β-aminomethyl-L-alanine (BAMA), β-*N*-methylamino-L-alanine (BMAA), and 2,4-diaminobutyric acid (DAB). Our objective was to investigate the impact of microbial diversity on NPAA production by cyanobacteria using semi-purified crude cyanobacterial cultures established from field samples collected by the Lake Winnipeg Research Consortium between 2016 and 2021. NPAAs were detected and quantified by ultra-performance liquid chromatography–tandem mass spectrometry (UPLC-MS/MS) using validated analytical methods, while Shannon and Simpson alpha diversity scores were determined from 16S rRNA metagenomic sequences. Alpha diversity in isolate cultures was significantly decreased compared to crude cyanobacterial cultures (*p* < 0.001), indicating successful semi-purification. BMAA and AEG concentrations were higher in crude compared to isolate cultures (*p* < 0.0001), and AEG concentrations were correlated to the alpha diversity in cultures (r = 0.554; *p* < 0.0001). BAMA concentrations were increased in isolate cultures (*p* < 0.05), while DAB concentrations were similar in crude and isolate cultures. These results demonstrate that microbial community complexity impacts NPAA production by cyanobacteria and related organisms.

## 1. Introduction

Harmful cyanobacterial blooms are a global public health concern and consistently affect Lake Winnipeg in Manitoba, Canada [1,2]. Lake Winnipeg is the tenth largest freshwater lake in the world with two distinct basins (north and south) separated by a narrow channel [3]. In the mid-1990s, an increased input flow from the Red River watershed incited an ecological state change of the lake [4,5,6]. Lake Winnipeg has been subsequently impacted by heightened agricultural, industrial, and urban activities, in addition to greater spring nutrient loads and rising global temperatures that have accelerated eutrophication [7,8,9]. These factors have resulted in increased cyanobacterial bloom frequency and severity [2,6,10,11,12,13]. The Lake Winnipeg Research Consortium Inc. (Gimli, MB, Canada) was founded in August 1998 to coordinate scientific research on the lake following the 1997 Red River flood [14].

One of the main concerns arising from Lake Winnipeg cyanobacterial blooms is the production of cyanobacterial toxins, including non-protein amino acids (NPAAs) which are associated with adverse health effects [1,15,16,17]. β-*N*-methylamino-L-alanine (BMAA) is a NPAA first identified in cycad seeds from Guam collected by the ethnobotanist Marjorie Grant Whiting [18,19,20]. Further investigation showed that the BMAA found in cycads was attributable to the presence of symbiotic cyanobacteria growing on cycad roots [21,22]. This resulted in the hypothesis that BMAA could bioaccumulate in ecosystems and food webs, thereby leading to a higher consumption by humans than previously thought [23]. Since then, this NPAA and its isomers 2,4-diaminobutyric acid (DAB), *N*-(2-aminoethyl)glycine (AEG), and β-aminomethyl-L-alanine (BAMA) have been detected in all five cyanobacterial sections from fresh water and marine ecosystems around the world [17,22,24]. More than 50 publications have documented the presence of BMAA in over 30 cyanobacterial genera since 2005; however, discrepancies in the literature have highlighted the importance of using validated and fit-for-purpose methods to analyze NPAAs in cyanobacterial matrices [25,26,27]. 

The presence of NPAAs in Lake Winnipeg is concerning due to the wide variety of neurotoxic effects they display, including misincorporation into proteins [21,28,29,30,31] and inducing oxidative stress in several neuronal cell lines through mechanisms including glutamate receptor excitotoxicity, cystine/glutamate antiporter inhibition, and modulation of the canonical Wnt signaling pathway [32,33,34,35,36,37]. Since these compounds are generally found in relatively low concentration in the environment, they may accumulate in tissues over time to produce neurotoxic effects [21]. The neurotoxicity of BMAA and DAB has been demonstrated with in vitro and in vivo models including rodents, birds, primates, and zebrafish [20,38,39,40]. In one study, vervets fed a low dose of BMAA (210 mg/kg/day) over 140 days developed neuropathy—including neurofibrillary tangles and β-amyloid deposits—consistent with neurodegeneration [40]. Therefore, understanding the factors that influence NPAA accumulations in areas with dense human populations that experience cyanobacterial blooms is necessary to mitigate health concerns due to the risk of long-term exposure.

Presently, the biosynthetic route to BMAA and other NPAAs in cyanobacteria is under investigation. NPAAs are likely ancient molecules and have been found in early earth geobiological samples such as meteorites and microbialites [41,42]. DAB is particularly abundant in nature and can be produced by several species of plants and microbes [41,43,44]. Recently, a bioinformatics approach was used to explore mechanisms for the 2,3-diaminopropanoic acid-dependent biosynthesis of BMAA in plants, bacteria, and cyanobacteria [44]. This approach was complicated by variations in analytical methods and cyanobacterial culture state of the data available; of the 28 cyanobacterial studies used, only 25% explicitly used all-axenic cultures, while others employed mono-cyanobacterial cultures, environmental samples, or did not specify. However, achieving axenicity in algal cultures is markedly difficult due to the presence of obligate symbionts and heterotrophic “helper” bacteria, and even with flow cytometry and 16S rRNA amplicon sequencing it is possible to miss contaminants [45]. As such, the impact of contaminating microbial species on NPAA biosynthesis in cyanobacterial cultures is unknown. 

Our previous collaboration with the Lake Winnipeg Research Consortium characterized the concentration profiles of NPAAs by cyanobacteria and related organisms in field samples collected in 2016 from diverse sites around Lake Winnipeg [17]. The objective of the current study was to understand how microbial diversity affects NPAA production in non-axenic cyanobacterial cultures. We hypothesized that cyanobacterial cultures from Lake Winnipeg would produce different amounts of NPAAs pre- and post- attempted purification of bacterial contaminants. To investigate our hypothesis, we tracked culture quality during the course of semi-purification and used 16S rRNA amplicon sequencing to determine the metagenomic composition of each culture. We then used a validated analytical method to determine NPAA concentrations in crude and semi-purified cultures, and correlated these concentrations to the alpha diversity in the cultures [17,46]. The results of this study offer an insight into how microbial community composition impacts NPAA profiles in cyanobacteria.

## 2. Results

### 2.1. Establishment and Semi-Purification of Cyanobacterial Cultures from Lake Winnipeg

In total, 27 crude cyanobacterial cultures were established from the Lake Winnipeg collections from 2016–2019. Semi-purification of these cultures and additional samples collected in 2021 yielded 16 additional isolate cyanobacterial cultures. Visual observation of cyanobacterial cultures using an inverted microscope revealed mixed various cyanobacteria from the orders Synechococcales (individual spherical unicellular cells), Chroococcales (stacked colonies consisting of 2–4 oblong cells), and various plant and algae-like organisms distributed across the lake (Appendix A). Two of the samples were identified as *Prochlorococcus marinus*. The records of qualitative culture tracking (Appendix A) and metagenomic compositions (Appendix A) are provided in the Appendix A; classifications were identical at the bootstrap levels of 50, 80, and 95. Although complete axenicity was not achieved in all isolates, a quarter of our isolated cultures contained >95% cyanobacterial DNA by 16S rRNA amplicon sequencing, and the proportion of cyanobacterial operational taxonomic units (OTUs) was relatively greater in the isolated cyanobacteria compared to the crude cultures.

### 2.2. Distribution of NPAAs in Crude and Isolate Cyanobacterial Cultures

#### 2.2.1. Concentration Profiles of NPAAs across Lake Winnipeg

The NPAA concentration profiles across Lake Winnipeg varied by isomer, as well as the relative purity of the culture (Figure 1; Appendix A). All established cultures produced detectable levels of total AEG. The highest levels of AEG in crude cultures (681 ± 50 ng/g) were produced by cyanobacteria grown from collections at Station 9 and in isolate cyanobacterial cultures from Station 5 NS (248 ± 53 ng/g), with both locations in the south basin of the lake. BAMA was detected in 24 of 27 crude cultures (89%) and in 14 of 16 isolate cultures (88%). The highest levels of BAMA were found in crude cultures established from Station 7NS (83 ± 27 ng/g) in the south basin, and in the isolate cultures from Station W4 (188 ± 97 ng/g) in the north basin of the lake. BMAA was detected in 52% (14 of 27) of the crude cyanobacterial cultures and in 63% (10 of 16) of the isolate cultures. The highest levels of BMAA in crude cultures (484 ± 53 ng/g) were established from Station 3C in the south basin of the lake, and in isolate cultures from Station W1 (73 ± 35 ng/g) in the north basin of the lake. DAB was measured in all cyanobacterial cultures, with the highest levels in crude cultures (8.4 ± 0.55 µg/g) found in a collection from Station 20 in the north basin of the lake and isolated cultures from Station 5 NS in the south basin (18.7 ± 2.34 µg/g).

#### 2.2.2. Occurrence of NPAAs in Paired Crude–Isolate Cyanobacterial Cultures 

Four of the crude and isolate cultures were derived from the same parent field sample, corresponding to a cyanobacterial bloom sample collected in 2019, and samples from Station W9, Station 20, and Station 19 collected in 2018 (shown on the maps in Figure 1). The phylogenetic composition of the cultures pre- and post-semi-purification are shown in Figure 2. As expected, in all four crude–isolate pairs, the percentage of reads corresponding to cyanobacteria was relatively higher in the isolate cultures compared to the crude cultures. The four paired crude–isolate cultures were all putatively identified as Chroococcales via morphological inspection; samples Bloom 2019, Station W9 2018, and Station 20 2018 contained oblong cells forming stacks of 2–4, while sample Station 19 2018 formed three-dimensional clusters of spherical cells. Samples Bloom 2019 and Station W9 2018 appeared identical in size (approximately 5–6 microns), while the cells in sample Station 20 2018 were similar, albeit slightly larger in size (approximately 6–8 microns) (Appendix A). Furthermore, samples Bloom 2019 and Station W9 2018 were putatively identified as containing the same cyanobacterial species based on morphological inspection (Appendix A).

A comparison of NPAA concentration profiles between crude and isolate cultures showed that statistically higher levels of AEG were produced in crude cultures (average 138 ± 16 ng/g) compared to isolate cultures (average 43 ± 7.8 ng/g; *p* < 0.0001; Figure 3a). AEG levels were statistically higher in the crude cultures of all four crude–isolate pairs. BMAA concentrations were statistically higher in crude cultures (average 136 ± 18 ng/g) compared to isolate cultures (average 25 ± 5.3 ng/g; *p* < 0.0001; Figure 3c). BMAA was detected in three of the isolate cultures and two of the paired crude cultures, with a significant decrease in the isolate culture established from Station 20 in the north basin. Conversely, BAMA concentrations were statistically higher in isolate cultures (average 53 ± 9.7 ng/g) compared to crude cultures (average 31 ± 3.0 ng/g; *p* < 0.05; Figure 3b). The levels of BAMA were generally higher in the paired crude cultures compared to isolate cultures, but not statistically different. DAB was found at similar levels in isolate cultures (average 2.5 ± 0.7 µg/g) compared to crude cultures (average 2.7 ± 0.2 µg/g; Figure 3d). The levels of DAB were significantly lower for paired isolate cultures established from the bloom site in 2019, as well as Stations 19 and 20. 

### 2.3. Analysis of NPAA Profiles and Alpha Diversity in Cyanobacterial Cultures

At a general level, the alpha diversity of the isolate cultures was significantly decreased compared to the crude cultures (*p* < 0.0001; Figure 4a). We then compared the influence of Shannon diversity scores on NPAA concentrations in crude and isolate cultures via the Pearson correlation test and a two-way analysis of covariance (ANCOVA; α = 0.05). The Pearson correlation test revealed that AEG was moderately correlated with Shannon diversity (r = 0.554; *p* < 0.0001) and all other NPAAs were weakly correlated with Shannon diversity (r < 0.250). The two-way ANCOVA revealed a significant influence of both culture type (*p* < 0.0001) and Shannon diversity (*p* = 0.014) on AEG concentrations. Culture type also had a significant influence on BMAA concentrations (*p* = 0.026) and DAB concentrations (*p* = 0.004), but none of BAMA, BMAA, or DAB were significantly influenced by Shannon diversity scores. The ordination plot showing the beta diversity (Bray–Curtis distances) revealed clustering of the crude cultures, indicating similarity in species composition (Figure 4b). Additionally, the principal component analysis biplot of NPAA concentrations showed that AEG and BMAA were associated with crude cultures and a higher Shannon diversity, while BAMA and DAB were associated with isolate cultures and a lower Shannon diversity (Figure 4c). None of the plots showed specific patterns related to the lake section of the initial collection site (statistical ellipses not shown); rather, data clustering was associated with the culture type (Appendix A). 

## 3. Discussion

Recently, there has been growing interest in exploring the role of environmental factors in the production of BMAA and other NPAAs by cyanobacteria and organisms present in complex ecosystems [47]. The cultivation of axenic algal and cyanobacterial cultures presents many challenges [45], but our data indicate that our strategy of the qualitative visual tracking of culture purity corresponded appropriately with our metagenomic findings. This showed that relatively successful cyanobacterial purification was accomplished without the use of additional chemical or instrumental strategies. Previous studies have shown that NPAA production in aquatic organisms is species-specific [43,48]. While we were not able to identify the primary species associated with the production of NPAAs, the metagenomic analysis of paired crude–isolate cultures presented here suggests that the relative complexity of a mixed-species culture influences NPAA levels. Additionally, our beta diversity analysis showed clustering of crude cultures compared to isolate cultures, highlighting the basal microbial community present within Lake Winnipeg [49]. The paired crude–isolate cultures from the same collection location contained residual bacteria from the genera *Dietzia* spp. and *Alkalibacterium* spp., both of which are chemoorganotrophic species [50,51]. AEG, BAMA, and BMAA are not well-characterized bacteria, but some microbial species are known producers of DAB, which could explain the higher concentration of DAB detected in most samples compared to its isomers [43,44]. 

Additionally, our data provide insights into the effects of microbial community composition on the production of NPAAs. Comparison of these data to our previous work showed that concentrations of NPAAs in uncultured field samples were orders of magnitude higher than when these field samples were cultured in vitro [17]. However, the concentrations for all in vitro cultures in this study as well as our previous study were much more similar, suggesting that growing cyanobacteria in optimized laboratory conditions in a nutritionally replete growth medium may alter the ability of some cyanobacterial strains to produce detectable BMAA [52,53]. The concentrations of NPAAs in environmental and lab-cultured samples vary greatly between studies and these results are further complicated by the wide range of analytical methods used to detect these compounds in biological matrices [25,27]. A recent study indicated that BMAA concentrations in cyanobacterial strains isolated from Northern Polish water bodies ranged from not detected (ND) to 0.8 µg/g, while cultured cyanobacterial strains ranged from ND to 0.6 µg/g (dry weight (DW); total BMAA) [54]. Our findings fall within the range of these values, as total BMAA concentrations ranged from ND to 0.48 µg/g DW and furthermore indicate that BMAA production in cyanobacteria is not strongly correlated to the amount of contaminant bacteria in culture, as BMAA concentrations were not significantly influenced by the alpha diversity of the culture. This is in line with previously published in vitro diatom studies showing that the lack of culture axenicity did not significantly affect concentrations of BMAA and other NPAAs in these species [43].

Researchers are using both in vitro and field monitoring approaches to uncover the dynamics between cyanobacteria and other BMAA-producing organisms in complex microbial communities [55,56,57,58,59]. BMAA metabolism in cyanobacteria is hypothesized to be closely associated with environmental nitrogen levels and nitrogen fixation [24,60,61]. Many recent studies have examined the effects of nitrogen availability on NPAA production by cyanobacteria and related organisms using in vitro techniques or by tracking bloom samples over a season [24,60,61,62]. During non-favorable environmental conditions, such as a bloom outbreak or collapse, BMAA may accumulate and act as an antagonistic agent to provide a competitive advantage for non-nitrogen fixing cyanobacteria [15,63,64,65,66,67]. Additionally, certain microbial phyla may have evolved to co-exist with NPAA-producing organisms using specific mechanisms of protection. A new line of research shows that the gut microbiota of cycad-feeding insects, consisting primarily of Proteobacteria, produce iron-chelating metabolites that allow these insects to tolerate toxin-rich diets including BMAA [68].

In summary, the metabolic functions of NPAAs including BMAA have not been fully elucidated, and more research is needed to understand the role of these compounds in harmful algal blooms. Researchers are beginning to develop new in vitro approaches, such as co-culture systems and isotope tracing methods, to analyze metabolite exchange in multi-species microbial communities [69,70]. These emerging strategies could be used to trace the metabolic production pathways and exchange of NPAAs in controlled multi-species communities that include cyanobacteria and related organisms. These approaches would complement existing NPAA field monitoring strategies [61]. Elucidating the specific factors—including nutrient availability and antagonistic or mutualistic microbial interactions—that affect cyanotoxin production in multi-species communities where numerous cyanobacteria species co-exist in varying abundances will lead to the development of better mitigation strategies to reduce cyanotoxin exposures in areas experiencing harmful algal blooms. 

## 4. Materials and Methods

### 4.1. Sample Collection and Cyanobacterial Culture Establishment

#### 4.1.1. Field Sample Collection 

Samples were collected by the Lake Winnipeg Research Consortium in 50 mL conical centrifuge tubes (Corning CentriStar; Corning, New York, NY, USA) from the surface (0–0.5 m depth) of standard stations and from cyanobacterial bloom sites during the voyage of the research ship *Namao* in the summer and fall seasons of 2016–2019, and 2021. Two samples were collected from each location during the summer and fall voyages of the *Namao*. All samples were identified by the station or GPS location of collection. The samples were shipped at an ambient temperature for the establishment of in vitro cultures of the cyanobacteria. The samples were couriered to the University of British Columbia Okanagan within 1 week of the voyage. 

#### 4.1.2. Crude Cyanobacterial Culture Establishment 

Upon arrival to the University of British Columbia Okanagan, field samples were briefly vortexed to homogenize cells. Using sterile inoculation loops (Nunc™ Disposable Loops and Needles; Thermo Scientific, Waltham, MA, USA), 100 µL of cells from each sample were spread onto Petri dishes (VWR, Mississauga, ON, Canada) containing complete BG_11_ media (Sigma Aldrich, Oakville, ON, Canada) solidified with agar (Fisher Scientific, Ottawa, ON, Canada; 4 g agar/500 mL media) and autoclaved (Steris, Mentor, OH, USA; 121 °C at 18 psi for 20 min), as described previously [17]. Crude cultures were incubated in a controlled-environment growth room at 27 °C with a 16 h photoperiod (Sunblaster™ full spectrum daylight fluorescent bulbs; 40 μmol/m^2^/s) and subcultured monthly by streaking onto fresh BG_11_ agar plates using sterile inoculation loops. After five months of sequential culture, crude cyanobacterial cultures were transferred into bioreactor boxes (Liquid Lab Vessel; Caisson Laboratories, Smithfield, VA, USA), to which 50 mL liquid BG_11_ media were added. These boxes were incubated in the same controlled-environment room with regular replenishment of media to mitigate volume loss due to evaporation. 

#### 4.1.3. Isolate Cyanobacterial Culture Establishment

To generate isolate cyanobacterial cultures, crude cultures were screened at 3× magnification (Nikon SMZ745 stereo microscope, Brighton, MI, USA), and cyanobacterial colonies were subsequently isolated and cultured onto individual Petri dishes with sterile inoculation loops. Isolate cultures were incubated in the previously described controlled-environment room and were routinely monitored at 3× magnification for the presence of contaminating bacteria. The culture quality was ranked qualitatively on a scale of 1–5, with 1 indicating heavy contamination (i.e., no regions of uncontaminated cyanobacterial growth) and 5 meaning no visible contamination (Appendix A). When the culture growth appeared to plateau, indicated by no further increases in colony size, they were checked under the stereo microscope and clean regions were marked for subculturing to gradually purify the samples of contaminants.

### 4.2. Detection and Quantification of Non-Protein Amino Acids

#### 4.2.1. Sample Preparation

The total NPAA acid content in crude cultures was measured by pipetting approximately 1.5 mL of cells from each bioreactor box or Petri dish into a pre-weighed microcentrifuge tube (1.5 mL; Eppendorf, Hamburg, Germany), which was frozen at −20 °C. Frozen cells, including their extracellular matrices, were lyophilized to complete dryness (FreeZone 4.5; Labconco, Kansas City, MI, USA). Sample preparation, detection, and quantification followed the previously published validated analytical methods, with the data collection time extended to include BAMA [17,46]. In brief, approximately 10–20 mg of lyophilized cells was accurately weighed using an analytical balance (Ohaus; VWR, West Chester, PA, USA) and transferred into a glass hydrolysis vial (15 mm × 45 mm; Fisher Scientific). 1000 µL of 6 N HCl were added to the hydrolysis vial, purged with nitrogen gas for 30 s, then hydrolyzed at 110 °C for 18 h on a heating block (VWR Standard Dry Block Heater). After cooling, 400 µL of sample was pipetted into a centrifuge filter tube (0.2 µm nylon membrane; VWR) and centrifuged (VWR Galaxy 16DH centrifuge) for 5 min at 13,000 rpm (16,000× *g*). A 10 µL aliquot of the filtrate was placed into a polypropylene centrifuge tube (1.5 mL; Fisher Scientific) and dried to complete dryness in a Speedvac (Labconco Centrivap, VWR). The dried-down filtrate was then reconstituted with a borate buffer (pH 10) and derivatized (see Section 4.2.2). 

For separate measurement of protein-bound and free NPAA content in isolate cultures, approximately 10–20 mg of fresh cells were collected directly from the Petri dish into pre-weighed microcentrifuge tubes and analyzed without lyophilization. The free amino acid portion was extracted by pipetting 200 μL of 0.1 N TCA to each sample and vortex-mixing at a maximum speed for 30 s. The samples were then centrifuged for 5 min at 13,000 rpm and the supernatant was pipetted into a centrifuge filter tube and centrifuged for 5 min at 13,000 rpm to yield the free amino acid extract. The pellet was resuspended in 100 μL 0.1× phosphate-buffered saline and transferred to a glass hydrolysis vial; 900 μL of 6 N HCl was added to the vial and the rest of the procedure followed as previously stated to collect the protein-bound amino acid portion. 

#### 4.2.2. Derivatization Reaction and Quantification of NPAAs 

The sample filtrates were reconstituted with 80 µL a 0.2 M borate buffer and derivatized with 20 µL AccQ-Fluor reagent (Waters Corp., Mississauga, ON, Canada; reconstituted as per the kit instructions). Complete reaction was ensured by vortex-mixing (Vortex Genie 2; Scientific Industries, Bohemia, NY, USA) and incubation for 10 min at 55 °C prior to analysis. Lysine was also monitored during analysis to ensure complete derivatization [71]. Samples were quantified by comparison to authentic standards (β-*N*-methylamino-L-alanine: Sigma Aldrich, CAS No. 16012-55-8; *N*-(2-aminoethyl)glycine: Sigma Aldrich, CAS No. 24123-14-6; 2,4-diaminobutyric acid: TCI America, CAS No. 1883-09-06), and a custom synthesis of β-aminomethyl-L-alanine described in the supplemental section to [17], derivatized in the same manner for each batch of samples, as described previously. The 100 µL aliquot of derivatized sample was then transferred to an autosampler vial (2 mL amber glass with pre-slit Teflon-coated caps; Waters Corp.) fitted with a conical bottom spring insert (250 µL glass; Canadian Life Science, Peterborough, ON, Canada) for separation and detection. 

All samples were analyzed by UPLC-MS/MS (Waters Acquity I-Class BSM, SM-FTN, column heater/cooler; Waters Xevo TQ-S with ESCi probe, MassLynx v4.1) with electrospray ionization (ESI+) and multiple reaction monitoring (MRM) on a Waters Acquity UPLC BEH C_18_ (2.1 mm × 100 mm; 1.7 µm particle size) column, as described in detail previously. A gradient elution of 20 mM ammonium formate at pH of 5.0 adjusted with glacial acetic acid (Fisher) (Solvent A) was used, and methanol (Fisher, Optima™ LC/MS grade) (Solvent B) with initial conditions of 90:10, reaching 50:50 at 7.00 min, 25:75 at 7.50 min and re-equilibrating to 90:10 at 9.60 min to 12 min before the next injection. The needle and seal wash solvent consisted of 90% acetonitrile (Fisher, Optima™ LC/MS grade) and 10% water (Millipore Ultrapure, 18.2 MΩ cm). The method developed by Glover et al. (2015) was fully validated by a Single Laboratory Validation protocol and there was no significant difference in the performance characteristics of the current analysis. Multiple reaction monitoring (MRM) transitions and other mass spectrometry parameters are presented in Appendix A. The complete baseline separation of all analytes was achieved, ensuring accurate quantification of each analyte. All standard curves had excellent linearity, with R^2^ > 0.99 in the linear dynamic range (0.37–1530 pg on column for each analyte). The method detection limit (MDL) was calculated by the U.S. Environmental Protection Agency guidelines, defined as the minimum measured concentration of a substance that can be reported with 99% confidence that the measured concentration is distinguishable from the method blank results [72]. The MDLs using the spiked blank method for AEG, BAMA, BMAA, and DAB were determined to be 0.34, 0.67, 0.48, and 1.19 pg on the column, respectively. The limits of quantification (LOQ) for AEG, BAMA, BMAA, and DAB were 1.17, 2.33, 1.67, and 4.11 pg on the column, respectively. The MDL for BMAA in a cyanobacterial matrix (*Nostoc* sp.) was 5.82 ng/g (dry weight). Due to endogenously high levels of AEG, BAMA, and DAB in this matrix, producing a signal-to-noise ratio of >20 in the samples, we could not calculate the matrix MDL for those analytes. 

### 4.3. Taxonomic Analysis of Cyanobacterial Cultures and Data Visualization

#### 4.3.1. Microscopic Images and Visual Identification of Cyanobacterial Orders 

Images were acquired using an inverted microscope (Olympus IX83; Olympus Canada Inc., Toronto, ON, Canada) at 40× magnification. The transmission images of the crude and isolate cultures were obtained using full color DIC (Differential Interference Contrast), while fluorescence images of the crude cultures were obtained by a non-staining epifluorescence method with an mCherry dichroic filter cube (excitation: 562 ± 20 nm, emission: 641 ± 37.5 nm). Microscopic images were edited using open-source ImageJ software (v1.51). The dominant cyanobacterial order/s present within each isolate culture were putatively identified using the morphological characteristics described in [73], the Cyanobacterial Key developed by Robin A. Matthews and Geoffrey B. Matthews at Western Washington University in 2018 (http://snoringcat.net/cyanobacteria_key/index.html, accessed on 25 April 2023), the cyanobacterial gallery maintained by Dr. Jason Oyadomari at Finlandia University (https://www.keweenawalgae.mtu.edu/index.htm, accessed on 25 April 2023), and the taxonomic key in the 2018 “A Guide to Cyanobacteria; Identification and Impact” [74]. 

#### 4.3.2. Next-Generation Sequencing (NGS) of Cyanobacterial Cultures

##### DNA Extraction

Cyanobacterial genomic DNA was extracted from the cultures using the Soil DNA Isolation Kit (Norgen Biotek #64000, Thorold, Schmon Pkwy, Canada) with some additional modifications to increase disruption of the cyanobacterial membranes. The frozen cell pellets were resuspended in 750 μL kit-provided Lysis Buffer G and vortexed for several seconds; the resuspensions were then transferred to kit-provided bead tubes. Of the kit-provided Lysis Additive A, 200 µL and 50 μL of 0.2 μg/μL lysozyme were then added, and the samples were vortexed at a maximum speed for 10 min, followed by sonication for 5 min. The samples were centrifuged (16,000× *g*, 2 min) and the supernatant transferred to a fresh tube. Of the Binding Buffer I, 100 µL was added, and the tubes were inverted 8× and incubated at 4 °C for 5 min. They were centrifuged (16,000× *g*, 4 min) and the supernatant again transferred. Of the OSR solution, 50 µL was added and the tubes were inverted 8× and incubated at 4 °C for 5 min. They were centrifuged (16,000× *g*, 4 min), and 700 μL of the supernatant was transferred to fresh tubes. Of the Lysis Buffer QP, 400 µL and 550 μL of ethanol were added, and the samples were vortexed and then passed through kit-provided spin columns (3 × 11,000× *g*, 30 s). Of the Binding Buffer B, 550 µL was passed through the spin columns (11,000× *g*, 1 min), then they were washed with 500 μL Wash Solution A (2 × 11,000× *g*, 1 min). The columns were dried by centrifugation (16,000× *g*, 4 min) and the columns transferred to the elution tubes, to which 50 μL Elution Buffer B was added. The elution tubes were incubated for 1 min at room temperature, and finally the genomic DNA was collected by elution (11,000× *g*, 1 min). Eluted gDNA was stored at −20 °C until it was sent for next-generation sequencing (NGS) analysis.

##### 16S rRNA Sequencing

Illumina MiSeq 16S rRNA sequencing of the samples was performed at the Sequencing and Bioinformatics Consortium at the University of British Columbia (Vancouver, BC, Canada). A 25 μL aliquot of each gDNA sample was submitted. The V3 and V4 regions of the 16S rRNA genes were PCR-amplified using literature-derived bacterial primers and then modified with Illumina sequencing adaptors and dual-index barcodes with the Nextera XT DNA Index kit. Libraries were then pooled and sequenced on a MiSeq Nano flow cell (Illumina, San Diego, CA, USA) to generate paired-end 250 bp reads. Illumina MiSeq Reporter (v2.6.2.3) was used to generate bioinformatic reports with taxonomic information for each sample.

##### DADA2 Taxonomic Identification

DADA2 (v1.16) was used to process the Illumina paired-end fastq files [75]. In brief, the read qualities of the samples were inspected and filtered; the first 20 and final 10 nucleotides from each sequence were trimmed to remove primer sequences and poor-quality regions. The forward and reverse reads were then merged, and chimeras removed. Taxonomy was assigned using CyanoSeq (v1.1.2), consisting of a curated cyanobacterial 16S rRNA database containing 5411 cyanobacterial sequences [76] and SILVA (v138.1), and a database containing more than 2 million 16S/18S rRNA bacterial, archaiac, and eukaryotic sequences [77]. Taxonomic assignments were performed at the bootstrap levels of 50, 80, and 95. Shannon and Simpson indexes of alpha diversity, and Bray–Curtis distances for beta diversity were calculated using phyloseq (v1.44) [78].

#### 4.3.3. Statistical Analysis and Visualization

Quantification, statistical analysis, and visualization of NPAA data was performed using Microsoft Excel^®^, R (v4.3.1) in RStudio^®^, GraphPad Prism (v9.5.1), and Adobe^®^, Illustrator (v27.5). Data from the chromatographic analysis of NPAAs were collated in Excel^®^ to calculate the concentrations of each isomer in each sample. All averaged concentration data are presented with the standard error of the mean. For statistical purposes, undetected (ND) values were replaced with a value of 1/10th of the method detection limit for each compound and back-calculated to units of ng/g based on the starting sample weight and sample dilution scheme. As none of the raw concentration values for NPAAs followed a normal distribution by the D’Agostino and Pearson test, we log-transformed all values before performing the Pearson correlation test and a two-way analysis of covariance (ANCOVA; α = 0.05). After the log-transformation, all concentration values followed a normal distribution except for BMAA due to the high number of samples with an ND value. The statistical tests used for each analysis are described in the figure captions. A site map and GPS coordinates of the collection sites used for this study were provided by the Lake Winnipeg Research Consortium. NPAA concentrations were site-mapped using the GeoJSON package (v0.3.4) from R [79] and the geomap provided by the University of Manitoba Centre for Earth Observation Science [80]. All other data were plotted using GraphPad Prism or the ggplot2 package [81] in R.

## Figures and Tables

**Figure 1 toxins-16-00169-f001:**
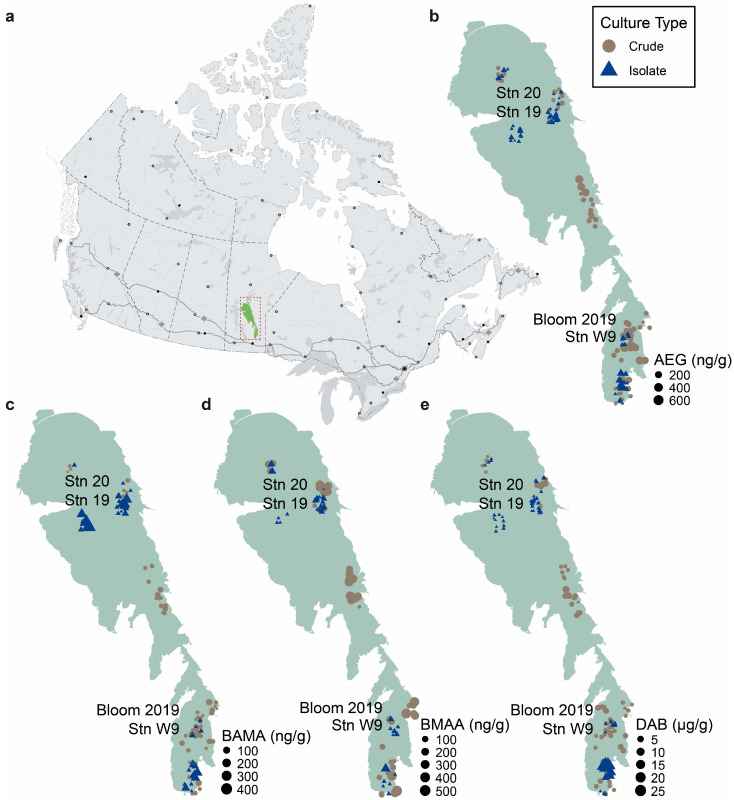
Concentration profiles and distribution of non-protein amino acids in Lake Winnipeg. (**a**) Lake Winnipeg (highlighted in the red box) is located in Manitoba, Canada. (**b**) *N*-(2-aminoethyl)glycine (AEG); (**c**) β-aminomethyl-L-alanine (BAMA); (**d**) β-*N*-methylamino-L-alanine (BMAA); and (**e**) 2,4-diaminobutyric acid (DAB) concentrations in crude (brown; *n* = 27 samples with 3 technical replicates) and isolate (blue; *n* = 16 samples with 4 technical replicates) cultures. Original field samples were collected between 2016 and 2021. Stations containing paired crude and isolate cultures from the same original field sample are shown on each map. Canada map outline adapted from the Atlas of Canada, Natural Resources Canada, and Lake Winnipeg outline adapted from the University of Manitoba Centre for Earth Observation Science.

**Figure 2 toxins-16-00169-f002:**
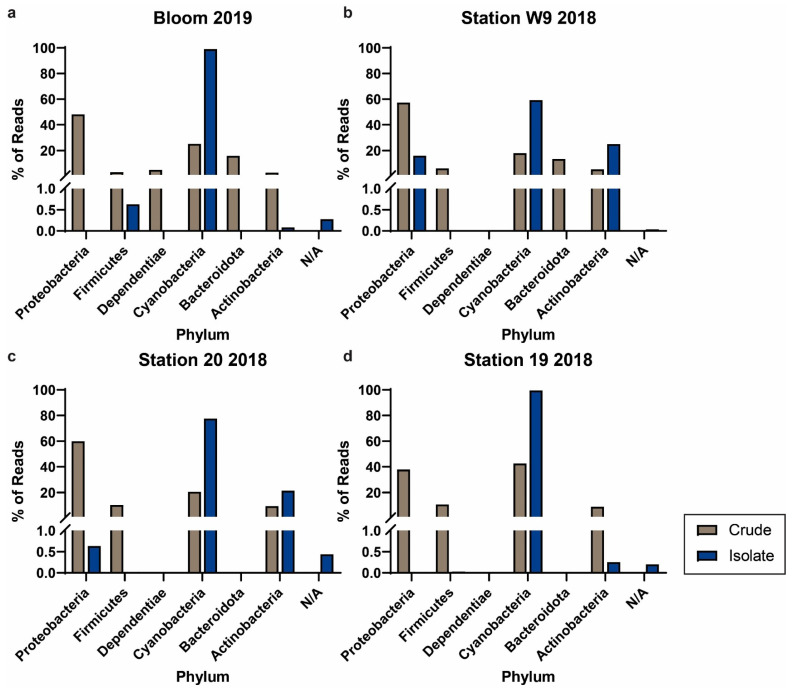
Phylum breakdown of paired crude (brown) and isolate (blue) samples from the same parent field samples. Cultures were established from locations and years (**a**) Bloom 2019, (**b**) Station W9 2018, (**c**) Station 20 2018, and (**d**) Station 19 2018. Cultures shown in (**a**) and (**b**) were putatively identified as containing identical cyanobacterial species.

**Figure 3 toxins-16-00169-f003:**
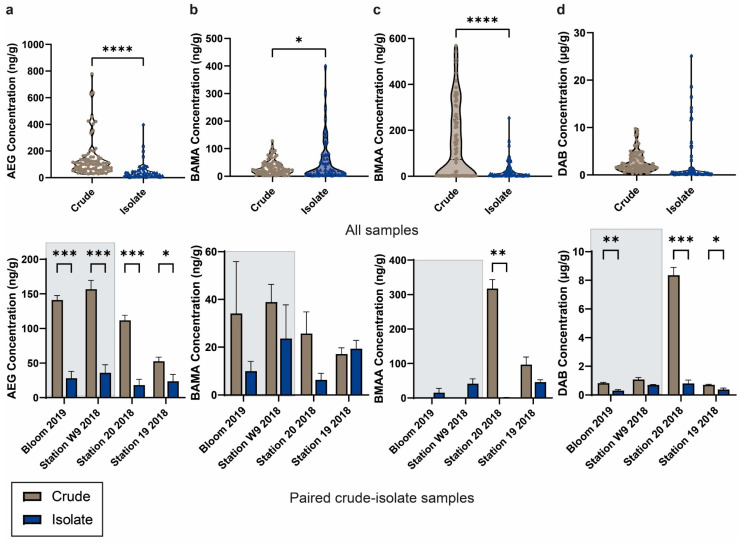
Comparison of non-protein amino acid concentrations in crude and isolate cultures. (**a**) *N*-(2-aminoethyl)glycine (AEG); (**b**) β-aminomethyl-L-alanine (BAMA); (**c**) β-*N*-methylamino-L-alanine (BMAA); and (**d**) 2,4-diaminobutyric acid (DAB) concentrations in all crude (brown; *n* = 27 samples with three technical replicates each, and isolate (blue; *n* = 16 samples with four technical replicates each) cultures and comparisons between paired crude and isolate cultures from the same original field sample. Error bars indicate standard error of the mean and not detected values were estimated as 1/10th of the method detection limit for the calculation of statistical differences via an unpaired *t*-test (α = 0.05) where * *p* < 0.05; ** *p* < 0.01; *** *p* < 0.001; **** *p* < 0.0001. Samples Bloom 2019 and Station W9 2018, highlighted in grey, were putatively identified as containing identical cyanobacterial species.

**Figure 4 toxins-16-00169-f004:**
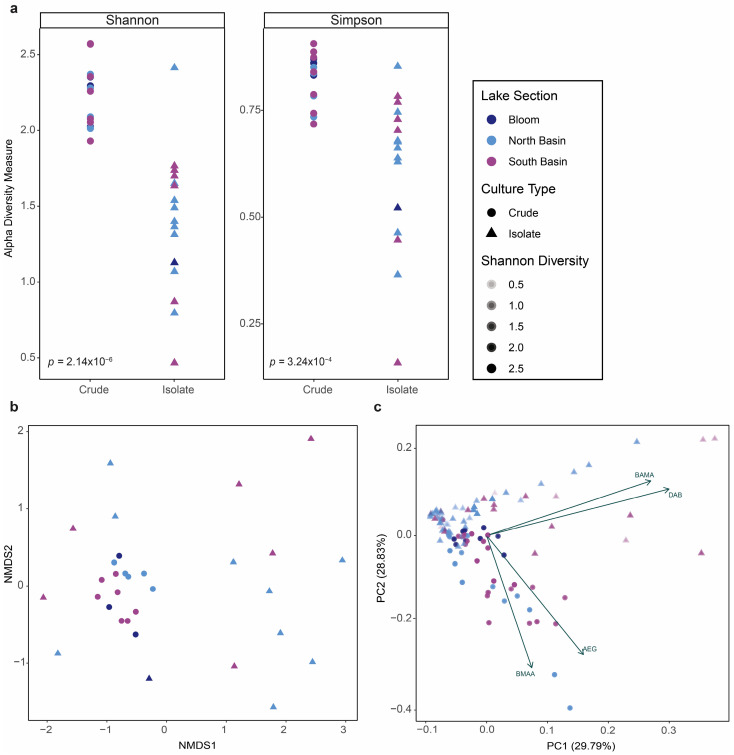
Correlations between microbial diversity and non-protein amino acid (NPAA) production in cultures. (**a**) Alpha diversity measures (Shannon, Simpson) of crude cultures (circles) compared to isolate cultures (triangles), colored according to the location of initial field collection site. Statistically significant differences were calculated by an unpaired *t*-test (α = 0.05). (**b**) Ordination (non-metric multi-dimensional scaling; NMDS) plot showing the beta diversity measure (Bray–Curtis distances) of crude cultures (circles) compared to isolate cultures (triangles), colored according to the location of initial field collection site. (**c**) Principal component analysis biplot of auto-scaled NPAA concentrations from crude cultures (circles) compared to isolate cultures (triangles), colored according to the location of initial field collection site and with alpha values (shading) showing the Shannon diversity scores. Undetected values were estimated as 1/10th of the method detection limit for statistical purposes.

## Data Availability

Summarized data are available in the Supporting Information or as a supplemental data file. Any additional data are available from the authors by request.

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
