# Peer review of "Microbial Diversity Impacts Non-Protein Amino Acid Production in Cyanobacterial Bloom Cultures Collected from Lake Winnipeg"

_toxins, 2024, doi:10.3390/toxins16040169_

Round 1

Reviewer 1 Report

Comments and Suggestions for Authors

The manuscript “Microbial diversity impacts non-protein amino acid production in cyanobacterial bloom cultures collected from Lake Winnipeg”, follows the impact of microbial diversity on non-protein amino acids (NPAA) production by cyanobacteria using semi-purified crude cyanobacterial cultures established from field samples collected by the Lake Winnipeg.

In my opinion, the topic is very interesting since there is no detailed information on NPAA in cyanobacteria, appropriate methods were used and they present appropriate material and techniques to recreate the experiments and analyze the data. The aim and the hypothesis are stated in a way clear and concise and the results show excellent, methodical and detailed research work.

In general, I do find this manuscript well written; the authors might consider a final proofing paying special attention to the enchainment of ideas in some sections but I do not consider it is a reviewer's duty to care for these. Like from line 90 to 96 I consider eliminating the paragraph or taking it towards methodology. In Mat and Meth, place a first paragraph with the location in coordinates of Lake Winnipeg or a map of Canada to facilitate the location in reading and not have to look for the complementary material or the cited reference and finally update some references from 2008/2009.

As I do not have English as my native language, I do not know if I am in a position to correct it exhaustively, but I can observe a fluent and consistent reading.

Author Response

Thank you for your helpful comments on our manuscript. We have now included an outline map of Canada with Lake Winnipeg highlighted in Figure 1 to orient the reader to its location.

Thank you for your suggestion to consolidate our introduction by removing lines 90 to 96. However, we decided it would be necessary to orient the reader to the differences in objectives between this study and our previous study of field samples in Lake Winnipeg, so we left those sentences in the introduction.

In response to your suggestion to update references from 2008/2009, we have tried to include both seminal early papers that discuss NPAA production and neurotoxicity including Lobner et. al. (2007), Esterhuizen et. al. (2008), and Liu et. al. (2009) as well as more recent papers that discuss the same phenomena from the last 8 years. We are happy to include any other important references the reviewer believes we excluded.    

Reviewer 2 Report

Comments and Suggestions for Authors

Dear Authors,

You presented a huge amount of experimental data in your manuscript, but your conclusions are not clear enough.  

I hope that by answering the following questions and accepting my recommendations, your manuscript will be improved.

1. In the introduction you wrote that you aimed to "investigate the impact of microbial diversity on NPAA production by cyanobacteria". Your conclusions did not meet this objective. It is not clear which bacterial group/s could be related to the synthesis of BMAA, AEG, DAB or BAMA. How do your metagenomic data confirm that NPAA production is species-specific? – line 220, page 7

2. According to the literature and your manuscript, NPAAs are produced by cyanobacterial genera, but your isolated cultures do not show high values of these components. For example, in the Snt3C_2017_crude sample you found the highest value of BMAA and only 16.16 ng/g in the same sample, isolate 1. This approach seems to be only suitable for BAMA monitoring. Please discuss these results.

3. It is not possible to trace the dynamics of NPPAD measured during the period of your study - 2017 -2021. It would be very useful to demonstrate the content of BMAA in 2021 in the crude sample for example. Same for other compounds?

4. You did not discuss the concentration of compounds you measured in all samples during your study. Is the value 484 ± 53 ng/g BMAA too high or too low? Please compare your data with the published literature.

5. Also, the values presented in Table S1 are from both crude samples and isolate samples, but you must specify the starting material. According to the material and methods, dry material was used for the NPPAD extraction from crude samples and fresh material for the extraction from isolate samples.

6. One technical error - the paragraphs 2.2.1. and 2.2.2. have the same titles?

Comments on the Quality of English Language

Minor editing of English language required

Author Response

Thank you for your careful review of our manuscript and we have incorporated your recommendations as follows, which we believe have improved our manuscript:

  1. Thank you for pointing out the ambiguity in this statement. We have corrected this sentence as follows:

Previous studies have shown that NPAA production in aquatic organisms is species specific [44,49]. While we were not able to identify the primary species associated with the production of NPAAs, the metagenomic analysis of paired crude-isolate cultures presented here suggests that the relative complexity of a mixed-species culture influences NPAA levels.

Additionally, we have improved our analysis of alpha diversity in crude and isolate cultures (Section 2.3), which will hopefully clarify this statement.

  1. Thank you for bringing up this important point of consideration. The crude sample collected in 2017 from Stn 3C did contain the highest level of BMAA in our study. However, the isolate sample containing 16.16 ng/g BMAA was collected from a different location in Lake Winnipeg (Stn 1) in 2017 so direct comparison between these two samples would be challenging. While cyanobacteria are known to produce NPAAs, this study along with a growing body of literature suggests that other organisms may also produce NPAAs. Additionally, we hypothesized that cyanobacteria may produce NPAAs in response to environmental factors including other microbial species present and therefore, we would expect to see some difference in NPAA in crude and isolate cultures. More studies are needed to determine the exact metabolic function of NPAA production in cyanobacteria, and we have expanded on this point in the discussion.
  2. Unfortunately, we only performed the analysis of crude samples between the collection years 2016 – 2019 and samples collected in 2021 were only analyzed after the process of cyanobacterial semi-purification (isolate cultures). Therefore, we do not have any NPAA data for crude cultures collected in 2021.
  3. Thank you for pointing this out. Reported concentrations of NPAAs vary greatly between studies and this is a highly discussed topic in the literature. We have added more discussion of this topic in the discussion section and included a recent study that looked at both field and cultured samples of cyanobacteria as a point of reference:

Concentrations of NPAAs in environmental and lab-cultured samples vary greatly be-tween studies and these results are further complicated by the wide range of analytical methods used to detect these compounds in biological matrices [25,27]. A recent study indicated that BMAA concentrations in cyanobacterial strains isolated from Northern Polish water bodies ranged from not detected (ND) to 0.8 µg/g while cultured cyanobac-terial strains ranged from ND to 0.6 µg/g (dry weight (DW); total BMAA) [55]. Our find-ings fall within the range of these values, as total BMAA concentrations ranged from ND to 0.48 µg/g DW and furthermore indicate that BMAA production in cyanobacteria is not strongly correlated to the amount of contaminant bacteria in culture, as BMAA concen-trations were not significantly influenced by alpha diversity of the culture.

  1. Thank you for pointing out this omission and we have clarified the starting material for each type of culture in the Table S1 description.

Thank you for pointing out this error and it has been corrected.

Reviewer 3 Report

Comments and Suggestions for Authors

Overall, the article presents valuable insights into the relationship between microbial diversity and the production of neurotoxic non-protein amino acids (NPAAs) by cyanobacteria in Lake Winnipeg. However, several revisions are necessary to enhance the clarity, robustness, and conclusiveness of the findings. Below are the major points for revision.

The article lacks information regarding the normality check of the data, which is crucial when performing statistical operations like Pearson correlation.

The significance of correlations, particularly when dealing with small effect sizes (e.g., r=0.02), needs to be addressed, e.g. “…weak negative correlation to alpha diversity (r < -0.015)”. Discussing the practical implications or potential biological significance of these correlations would provide context for the readers.

Incorporating the interaction of effects, such as in a two-way ANOVA module, could enrich the analysis and support assumptions about the linkage between NPAA concentrations in cyanobacterial bloom cultures and the complexity of microbial communities present in mixed species cultures.

Expanding upon future research directions is also crucial, particularly concerning the metabolic functions of NPAAs and their implications for cyanotoxin exposure and mitigation strategies. This would help advance our understanding of cyanobacterial bloom dynamics and contribute to the development of effective mitigation approaches

Comments on the Quality of English Language

Check for grammatical errors, typos, and clarity issues throughout the manuscript to enhance readability and professionalism

Author Response

Thank you for your careful review of our manuscript and we believe the improved analyses we implemented based on your recommendations have greatly improved the manuscript. The normality check of the data revealed that the concentration data for all NPAAs were not normally distributed, so we decided to perform a log-transformation of the data before performing the Pearson correlation test. The updated values and description of the normality check are now included in the results and methods section. According to your suggestion, we also performed an analysis of covariance (ANCOVA) on NPAA concentrations with the independent variable culture type and covariant Shannon diversity score. These results are now included in the results section 2.3.

We have also improved our discussion of the implications or potential biological significance of these correlations. Reported concentrations of NPAAs vary greatly between studies and this is a highly discussed topic in the literature. We have added more discussion of this topic in the discussion section and included a recent study that looked at both field and cultured samples of cyanobacteria as a point of reference:

Concentrations of NPAAs in environmental and lab-cultured samples vary greatly be-tween studies and these results are further complicated by the wide range of analytical methods used to detect these compounds in biological matrices [25,27]. A recent study indicated that BMAA concentrations in cyanobacterial strains isolated from Northern Polish water bodies ranged from not detected (ND) to 0.8 µg/g while cultured cyanobac-terial strains ranged from ND to 0.6 µg/g (dry weight (DW); total BMAA) [55]. Our find-ings fall within the range of these values, as total BMAA concentrations ranged from ND to 0.48 µg/g DW and furthermore indicate that BMAA production in cyanobacteria is not strongly correlated to the amount of contaminant bacteria in culture, as BMAA concen-trations were not significantly influenced by alpha diversity of the culture.

We have also expanded on our discussion of the importance of elucidating the metabolic functions of NPAAs and future research strategies:

In summary, the metabolic functions of NPAAs including BMAA have not been fully elucidated and more research is needed to understand the role of these compounds in harmful algal blooms. Researchers are beginning to develop new in vitro approaches including co-culture systems and isotope tracing methods to analyze metabolite ex-change in multi-species microbial communities [72,73]. These emerging strategies could be used to trace the metabolic production pathways and exchange of NPAAs in con-trolled multi-species communities including cyanobacteria and related organisms, which would complement existing NPAA field monitoring strategies [64]. Elucidating the spe-cific factors –  including nutrient availability and antagonistic/mutualistic microbial in-teractions – affecting cyanotoxin production in multi-species communities where nu-merous cyanobacteria species co-exist in varying abundances will lead to the develop-ment of better mitigation strategies to reduce cyanotoxin exposures in areas experiencing harmful algal blooms.

Round 2

Reviewer 2 Report

Comments and Suggestions for Authors

Dear Authors,

Thank for your quick response. I'm satisfied with your answers and additional comments related to my questions.

Reviewer 3 Report

Comments and Suggestions for Authors

The authors thoroughly addressed all comments and reviewed the  article pretty good. The article would be published in the present form